# An adaptive extended radial basis function based interval analysis method for structural engineering solutions

Xu Jitang[1]*, He Jinli[2], Chen Qiang[3]

1 Department of Transportation and Construction, Tianjin Bohai Vocational Technical Collage, Tianjin, China, 2 Tianjin Bohai Vocational Technical College, Tianjin, China, 3 Beijing Zhongyu Shenzhou Data Technology Co., LTD, Beijing, China

* giftzhizhuo@gmail.com

## Abstract

In engineering structures, uncertain interval analysis is a study hotspot. However, how to obtain the accurate and efficient upper and lower solution bounds of uncertainty in the uncertain interval analysis is one of the key problems at present. To address this issue, the paper proposes a new adaptive extended radial basis function based interval analysis method to obtain accurate and efficient structural bound solutions. Instead of the pre-selecting the width of the basis functions, an Extended Radial Basis Function (E-RBF) and the adjusted widths are combined to confirm each width of the Gaussian basis function. In order to obtain the widths of the Gaussian basis function, the adaptive scaling technique is introduced in the E-RBF model. Then six numerical functions are selected to evaluate the accuracy of the E-RBF method with adjusted widths. On the basis of ensuring the accuracy of the widths of the Gaussian basis function, the adaptive E-RBF based interval analysis method is constructed by using the E-RBF with adjusted widths, catching the critical points and establishing the adaptive procedure for solving the extreme certain interval bounds of complex structural engineering problems. Next, three structural numerical examples and one composite shell are produced to verify the validity and effectiveness of this adaptive E-RBF based interval analysis method.

## Introduction

In most real-world engineering issues, uncertainty is ubiquitous such as uncertain environmental conditions, uncertain manufacturing errors, uncertain physical imperfections and so forth [1–4]. To enhance the credibility of these uncertain problems, the uncertainties that can be measured are classified into two types: the aleatory uncertainty and the epistemic uncertainty [5,6]. For aleatory uncertainty, the probabilistic method has been successfully applied to describe the inherent variation in an environment or system's behavior. In the probabilistic method, the precise

**Data availability statement:** All relevant data are within the manuscript.

**Funding:** Beichuang Teaching Assistant Program 2021BCE02010.

**Competing interests:** The authors have declared that no competing interests exist.

probabilistic distribution function with respect to the uncertain factors is required [7–11]. Dzhafarov [12] explained how all logically probable contextual influences can be quantified and classified with the help of probabilistic factors. Knoops [13] proposed a novel probabilistic finite element model for the prediction of postoperative facial soft tissues. Then a new unified framework was introduced for building probabilistic forecasting models for intermittent demand time series by Ali Caner [14]. Belay [15] presented a method for solving multiple-objective probabilistic fractional programming problem with discrete random variables.

However, this accurate function necessitates a great amount of statistical information for experimental data or simulation data, and it is sometimes less practical to obtain this probabilistic distribution function in the early stage of the design process [11,16–19]. As a result, the epistemic uncertainty method has been proposed to provide attractive alternative tools for engineering designs when the information data of uncertain factors is not sufficient or difficult to develop the probabilistic distribution function precisely. In the epistemic uncertainty research, non-probabilistic method has become a hot topic such as the fuzzy-set method [16,18,20–24] and the interval method [23,25–30]. Elishakoff and Ben-Haim may be the first to propose this non-probabilistic concept [31–33], and this method is considered a supplement to the traditional probabilistic method [9,34,35]. Specifically, the interval method or interval analysis may be considered the most widely utilized method or analytic tool [18,36–39]. I. Kim [40] proposed a censored time interval analysis to address the real-world censored time-to-event dataset of medical history. Y. Wang [41] combined the triangular fuzzy number evaluation index and Genetic-Tabu Search algorithm into the novel interval analysis the Non-Strict Uncapacitated Multi-Allocation p-hub Median Problem. Interval analysis is an effective solution of modeling or analyzing uncertain interval variables [42]. And it is easy to attain the lower and upper bounds of the design objectives for interval analysis considering the limited information under uncertain factors [29,39]. Therefore, this interval analysis, which treats uncertain variables as interval numbers for searching the upper and lower bounds, is much easier than the aleatory uncertainty or the probabilistic method.

Qiu and Elishakoff [43] proposed a generalization of the interval analysis for the non-probabilistic treatment of uncertainty using the anti-optimization technique. An optimum design of the uncertain mechanical system was presented based on interval analysis for the prediction of system response by S. Rao [44] and S. Chen [45]. C. Jiang [46–49] may be the first to propose a nonlinear interval number programming method and an interval analysis method for the calculation of the interval objective function caused by uncertainty. Unfortunately, the mentioned reviews can only solve problems with tiny interval ranges [10,27,50–52]. In other words, the traditional interval analysis cannot obtain accurate bounds when the uncertain deviation of interval parameters is large. Thus, a nested-loop or double-loop optimization-based process was developed to solve this unpleasant situation.

However, double-loop or nested-loop optimization-based process means more intensive computation. In order to save the computational cost, the approximation or surrogate model should be introduced for the nested-loop or double-loop interval analysis.

M. Abdellatief [53] presented a machine-learning technique to address the limitations of the complex, non-linear interactions between different mixing design parameters. To balance accuracy and efficiency, there is artificial intelligence method to predict the early-age compressive strength for optimal mix design [54]. And these reviews are the motivation of the E-RBF method. F. Li [39] proposed a double-loop multi-objective optimization based on the response surface method (RSM) approximation and sequential quadratic programming. J. Cheng constructed a nested-loop optimization using the radial basis function (RBF) [30] and the Kriging [55] approximation. J. Wu [3,29,51,56–58] also formulated a double-loop optimization to compress the overestimation of interval arithmetic with the Chebyshev surrogate model. J. Zhang [59] introduced a hybrid reliability analysis with both random and interval variables combining the projection outline-based active learning and Kriging metamodel to estimate the lower and upper bounds precisely. Also, a reliable and effective double-loop interval optimization was performed using the progressive trigonometric mixed response surface method by G. Zheng [60].

Nevertheless, there are two main shortcomings of the nested-loop or double-loop interval analysis with the approximation or surrogate model. One is the accuracy of the approximation or surrogate model. If more precise bounds are obtained, more sample data are needed [60]. The other disadvantage is the time-consuming in computations of the interval arithmetic, which means the nested-loop or double-loop optimization procedure will cost too much time and computations for the inner optimizations [29].

For these mentioned shortcomings, Z. Zhao [61] proposed a sequence of approximation techniques to guarantee the accuracy of the bounds by reconstructing the RBF model with densified samples in iterations. F. Li [10] established an adaptive Kriging model to improve the accuracy in the approximation of design functions. Y. Wu [62] presented a novel adaptive linear and normalized combination method for the component RBF to implement better function approximation and regression. J. Cheng [2] employed an adaptive Kriging model to compute dynamic characteristics efficiently. Z. Xing utilized an adaptive Metropolis-Markov Chain Monte Carlo method to assign the weights to the Kriging, RBF and the least squares support vector machines models. From above reviews, it can be proved that the adaptive technique is a powerful tool for the accuracy and efficiency of the approximation of the surrogate model. Among these approximate models, G. Pan [63] proposed a new sequential optimization sampling method to attain extrema points of the metamodel through the critical point obtained from the gradient equitation of the RBF model. The results in Pan's work proved that the RBF approximate model with critical points can offer a helpful way to deal with the time and computation consumption for diverse and complex engineering applications.

Although the RBF with the adaptive scaling algorithm can attain sufficient conditions and good approximation, there may be a notable glaring deficiency named spurious local minima because RBF may yield an interpolating surface with respect to a given set of prescribed data [64]. To deliberately avoid the notion of unique solvability, A. Mullur [64,65] proposed a more effective developing method called extended radial basis function (E-RBF). Rooh ul Amin [66,67] proposed the E-RBF to estimate the unmodeled dynamics of the octorotor. Consequently, this paper will utilize the adaptive E-RBF model and the adaptive E-RBF based interval analysis to overcome these drawbacks mentioned above.

This paper proposed an adaptive E-RBF based interval analysis method for engineering problems. The rest of this paper will be organized as below. The Extended radial basis function section introduces the Extended radial basis function with the adjusted width of the Gaussian basis functions. And six numerical examples are also proposed to test the accuracy in this section. Then the Adaptive E-RBF based Interval Analysis section is conducted to obtain more accurate and efficient interval bounds. In Section: Comparison to contemporary work, three structural problems and one composite shell example are performed to test the validity and effectiveness of this novel interval analysis method. Last, the Conclusions section illustrates the conclusion.

## Extended radial basis function

### Traditional radial basis function

The use of RBF as an approximation or surrogate model was first proposed by Hardy [68]. Since then, RBF, as a successful model, has been applied to all kinds of engineering and other approximations. The traditional radial basis function

(RBF) is a three-layer feed-forward network model that can output the response with the given training data [69–71]. The output response of this network function is the linear combination of basis functions (See Fig 1) and can be given by:

$$\hat{f}(\boldsymbol{x}) = \sum_{i=1}^{m} \omega_i \phi_i(\boldsymbol{x}) = \sum_{i=1}^{m} \omega_i \phi_i(\|\boldsymbol{x} - \boldsymbol{x}_i\|) \tag{1}$$

where $\boldsymbol{x}$ is the vector of the design variable; $m$ indicates the number of sampling points; $\boldsymbol{x}_i$ is the vector value of design variables at the $i$th sampling point; $\phi_i(\boldsymbol{x})$ is the $i$th basis function; $\|\boldsymbol{x} - \boldsymbol{x}_i\|$ denotes the Euclidean norm between the vector of the design variable and the vector value at the $i$th sampling point; $\omega_i$ represents the weight of the $i$th basis function. To evaluate the weights of the basis functions, Equation (1) should yield an interpolative function under a set of sampling points.

$$\sum_{i=1}^{m} \omega_i \phi_i(\|\boldsymbol{x}_k - \boldsymbol{x}_i\|) = y(\boldsymbol{x}_k), \quad k = 1, 2, \cdots, m \tag{2}$$

where $\boldsymbol{x}_k$ is the vector for the $k$th sampling point. In matrix notation, the RBF output response can be written as:

$$[\boldsymbol{\Phi}] \cdot \{\boldsymbol{\omega}\} = \{\boldsymbol{Y}\} \tag{3}$$

where

$$[\boldsymbol{\Phi}] = \begin{bmatrix} \phi\|\boldsymbol{x}_1 - \boldsymbol{x}_1\| & \phi\|\boldsymbol{x}_1 - \boldsymbol{x}_2\| & \cdots & \phi\|\boldsymbol{x}_1 - \boldsymbol{x}_n\| \\ \phi\|\boldsymbol{x}_2 - \boldsymbol{x}_1\| & \phi\|\boldsymbol{x}_2 - \boldsymbol{x}_2\| & \cdots & \phi\|\boldsymbol{x}_2 - \boldsymbol{x}_n\| \\ \vdots & \vdots & \ddots & \vdots \\ \phi\|\boldsymbol{x}_m - \boldsymbol{x}_1\| & \phi\|\boldsymbol{x}_m - \boldsymbol{x}_2\| & \cdots & \phi\|\boldsymbol{x}_m - \boldsymbol{x}_n\| \end{bmatrix}$$

$$\{\boldsymbol{\omega}\} = [\omega_1, \omega_2, \cdots, \omega_m]^T, \quad \{\boldsymbol{Y}\} = [y(\boldsymbol{x}_1), y(\boldsymbol{x}_2), \cdots, y(\boldsymbol{x}_m)]^T$$

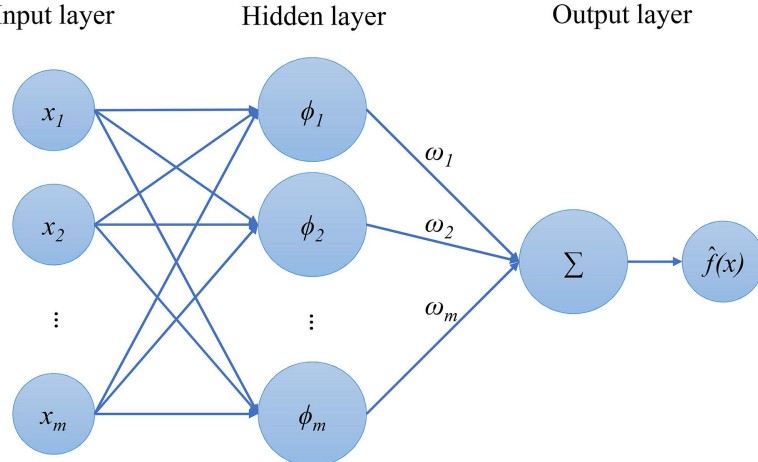

Input layer    Hidden layer    Output layer

**Fig 1. The traditional RBF diagram.**

Furthermore, the general basis functions are the cubic, Gaussian, multiquadric, inverse quadratic, and several other basis functions. The most commonly used basis functions are listed in Table 1, and c is the constant value in the basis function, which can be defined to obtain the best fit for the RBF model. From recent reviews, the Multiquadric and Gaussian functions can be considered the best-known and most often applied [63,72].

However, there is scarcely any simple way to evaluate the constant $c$ for basis functions [73]. In order to address this problem, S. Kitayama proposed an adjusted way of the width of the Gaussian basis function [69–71]. Hence, this paper adopts the Gaussian function as the basis function.

### Adaptive scaling technique

In order to determine the width of the Gaussian basis function in a simple manner, the following estimated width is introduced below:

$$c_i = \frac{d_{\max}}{\sqrt[n]{mn}}, \ i = 1, 2, \cdots, m \tag{4}$$

where $d_{\max}$ is the maximum distance between any two sampling points. $m$ is the number of sampling points, and $n$ is the number of design variables. It is obvious from Equation (4) that all the basis functions adopt the same constant value $c$. An improved simple estimate by Kitayama [70,71] is:

$$c_i = \frac{d_{i,\max}}{\sqrt{n} \cdot \sqrt[n]{m-1}}, \ i = 1, 2, \cdots, m \tag{5}$$

where $c_i$ indicates the width of the $i$ th Gaussian basis function, and $d_{i,\max}$ denotes the maximum distance between the $i$ th sampling point and the other sampling points. We can see from Equation (5) that each Gaussian basis function has its own individual width, which can deal with the non-uniform distribution of sampling points, unlike Equation (4).

In order to confirm the adjusted width of each basis function, a simple adaptive scaling technique is introduced here, and the scaled sampling points in the scaled space is:

$$x_{ij}^l = \frac{x_{ij} - x_{ij}^L}{x_{ij}^U - x_{ij}^L} \cdot s \ , i = 1, 2, \cdots, m \ ; \ j = 1, 2, \cdots, n \tag{6}$$

where $x_{ij}$ is the $i$th sampling point for the $j$th design variable. $x_{ij}^L$ and $x_{ij}^U$ represent the minimum (lower) and the maximum (upper) bounds for the $i$th sampling point the $j$th design variable, respectively. $s$ indicates the scaling coefficient for this adaptive scaling technique.

**Table 1. Commonly used basis functions.**

| Name | Basis function $d = \|x\text{-}x_i\|$ |
| --- | --- |
| Linear | $\phi((d)=cd$ |
| Cubic | $\phi(d)=(d+c)^3$ |
| Gaussian | $\phi(d)=exp(-d^2/c^2)$ |
| Multiquadric | $\phi((d)=(d^2+c^2)^{1/2}$ |
| Inverse quadratic | $\phi(d)=(d^2+c^2)^{-1}$ |

After this scaling transformation, the whole sampling points can be scaled between 0 and $s$ ($s>0$). In this adaptive scaling technique, the scaling coefficient $s$ plays a significant position and is also recommended as an adaptive value as follows:

$$s = \alpha \cdot s \ (\alpha > 1, s > 0) \tag{7}$$

where $\alpha$ is the adaptive coefficient for the scaling coefficient. By the experience of Kitayama, $\alpha = 1.2$ is suitable. All in all, the adaptive scaling technique can be shown in Fig 2 and be summarized as:

Step 1: The initial scaling coefficient $s$ should be set first;

Step 2: The whole sampling points are scaled according to Equation (6);

Step 3: The width of each basis function can be calculated following the scaled space (Step 2) based on the Equation (5);

Step 4: Search for the minimum width $r_{min}$ as below:

$$c_{min} = \min_{1 \leq i \leq m} \{c_i\} \tag{8}$$

Step 5: If $c_{min} \leq 1$, the scaling coefficient $s$ should be updated according to Equation (7), and repeat Step 2 to Step 5;

Otherwise, the adaptive scaling algorithm will be stopped.

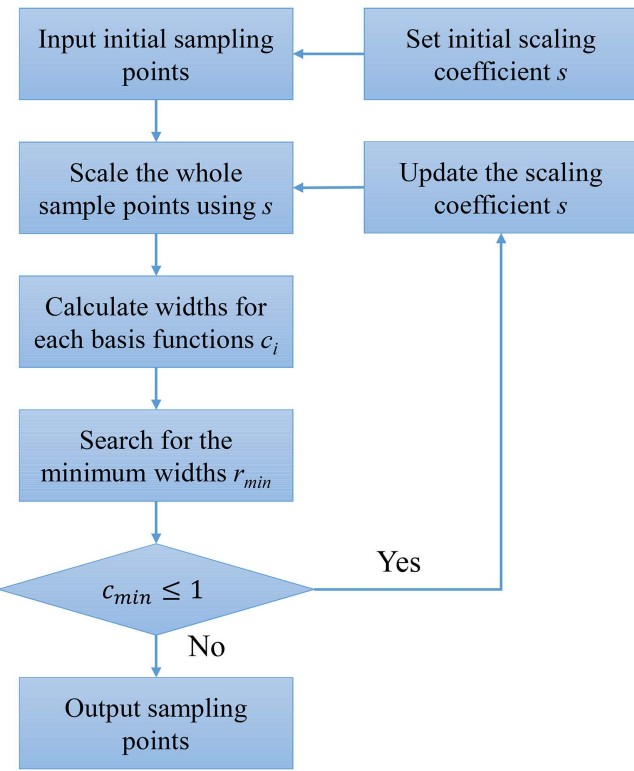

**Fig 2. The flowchart for the adaptive scaling technique.**

## Nonradial basis function

From the preceding discussion in the Introduction, the traditional RBF can only provide a unique interpolative solution and can not offer a means to express desirable properties. Thus, a so-called nonradial basis function is introduced to provide more freedom and flexibility to the traditional RBF. The nonradial basis function is not the function of the Euclidean distance $d$. Instead, the individual coordinates of a generic point with respect to a given sampling point are utilized in each dimension separately. Fig 3 displays the difference between the Euclidean distance $d$ and the relative coordinates $\xi$ for a two-dimensional space. Furthermore, the $n$-dimensional defined coordinate vector can be expressed as $\boldsymbol{\xi^i} = \boldsymbol{x} - \boldsymbol{x^i}$.

After the coordinates of the sampling points are obtained, the nonradial basis function for the $i$th sampling point and the $j$th dimension can be defined as $\psi_{ij}$, which can be constructed in three distinct components.

$$\psi_{ij}\left(\xi_i^j\right) = \alpha_{ij}^L \cdot \psi^L\left(\xi_i^j\right) + \alpha_{ij}^R \cdot \psi^R\left(\xi_i^j\right) + \alpha_{ij}^\beta \cdot \psi^\beta\left(\xi_i^j\right)$$

(9)

where $\alpha_{ij}^L$, $\alpha_{ij}^R$ and $\alpha_{ij}^\beta$ are coefficients of the nonradial basis functions to be determined for sampling points and given variables. The superscripts $L$ and $R$ denote the left and right sides of the nonradial basis functions. $\psi^L$, $\psi^R$ and $\psi^\beta$ are the nonradial basis functions, which can be evaluated in Table 2. The shape of the nonradial basis function in four distinct regions (I-IV) is depicted in Fig 4.

Table 2 and Fig 4 show that the proposed nonradial basis function forms differently in different regions (I-VI). The nonradial basis functions are $n$th-degree monomials plus a linear component in regions II and III ($-\gamma \le \xi_i^j \le \gamma$). While in regions I and VI ($\xi_i^j \le -\gamma, \xi_i^j \ge \gamma$), the nonradial basis functions are linear. The parameter $\gamma$ can be considered as a smoothness parameter. The suitable value of parameter $\gamma$ can depend on the magnitudes of the design problems. Generally, the better the smoothing properties are, the higher the parameter $\gamma$ is. The accuracy of the approximate model is not sensitive to the parameter $\gamma$. Thus, for normalization of the design space, the designer can choose parameter $\gamma = 0.25$. For most real-life engineering, the order $n$ should be 2 [64]. The complete nonradial basis function for the $i$th sampling point is the sum of the individual basis functions in each dimension:

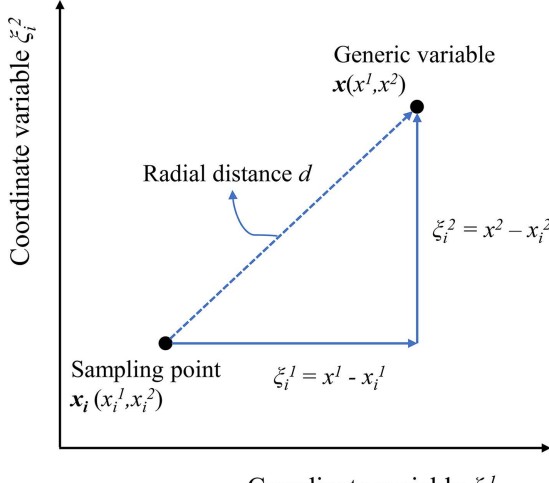

**Fig 3. The diagram of coordinates ξ.**

**Table 2. The nonradial basis function.**

| Region | Range of $\xi_i^j$ | $\psi^L$ | $\psi^R$ | $\psi^\beta$ |
|---|---|---|---|---|
| I | $\xi_i^j \leq -\gamma$ | $(-n \cdot \gamma^{n-1})\, \xi_i^j + \gamma^n(1-n)$ | $0$ | $\xi_i^j$ |
| II | $-\gamma \leq \xi_i^j \leq 0$ | $(\xi_i^j)^n$ | $0$ | $\xi_i^j$ |
| III | $0 \leq \xi_i^j \leq \gamma$ | $0$ | $(\xi_i^j)^n$ | $\xi_i^j$ |
| IV | $\xi_i^j \geq \gamma$ | $0$ | $0$ | $(n \cdot \gamma^{n-1})\, \xi_i^j + \gamma^n(1-n)$ | $\xi_i^j$ |

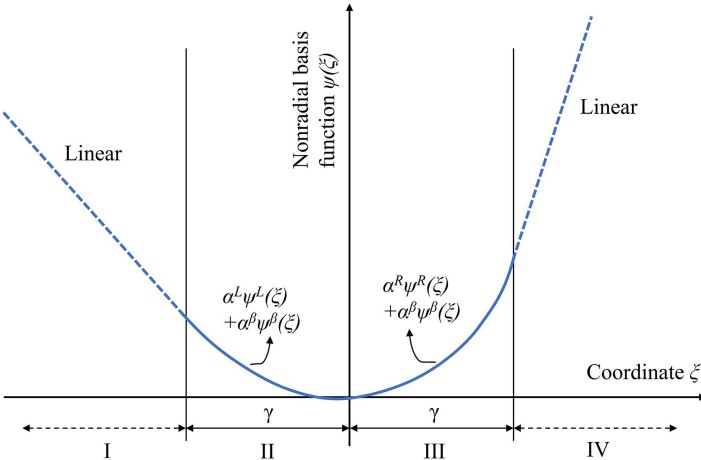

**Fig 4. The nonradial basis function diagram.**

$$\psi_i\left(\boldsymbol{x} - \boldsymbol{x}_i\right) = \psi_i\left(\xi_i\right) = \sum_{j=1}^{n} \psi_{ij}\left(\xi_i^j\right)$$

(10)

## Radial and nonradial basis function

In this section, the combination of radial and nonradial basis functions is introduced to attain the appealing properties of both types of basis functions. This combination approach is named extended radial basis function (E-RBF), which has the characteristics of effectiveness and flexibility. Using these two distinct basis functions, the E-RBF approximation is the linear combination of these two functions (see Fig 5), which can be defined as:

$$\tilde{f}\left(\boldsymbol{x}\right) = \sum_{i=1}^{m} \omega_i \phi_i\left(\|\boldsymbol{x} - \boldsymbol{x}_i\|\right) + \sum_{i=1}^{m} \psi_i\left(\boldsymbol{x} - \boldsymbol{x}_i\right) = \sum_{i=1}^{m} \omega_i \phi_i\left(d\right) + \sum_{i=1}^{m} \psi_i\left(\xi_i\right)$$

$$= \sum_{i=1}^{m} \omega_i \phi_i\left(d_i\right) + \sum_{i=1}^{m}\sum_{j=1}^{n}\left[\alpha_{ij}^L \cdot \psi^L\left(\xi_i^j\right) + \alpha_{ij}^R \cdot \psi^R\left(\xi_i^j\right) + \alpha_{ij}^\beta \cdot \psi^\beta\left(\xi_i^j\right)\right]$$

(11)

where $\phi_i\left(\|\boldsymbol{x} - \boldsymbol{x}_i\|\right)$ and $\psi_i\left(\boldsymbol{x} - \boldsymbol{x}_i\right)$ can be obtained in Table 1 and Equation (9), $\psi^L$, $\psi^R$ and $\psi^\beta$ can be found in Table 2. Then coefficients $\alpha_{ij}^L$, $\alpha_{ij}^R$ and $\alpha_{ij}^\beta$ can be defined as vector format $\alpha^L$, $\alpha^R$ and $\alpha^\beta$:

$$\alpha^L = \left[\alpha_{11}^L, \alpha_{12}^L, \cdots, \alpha_{1n}^L, \alpha_{21}^L, \cdots, \alpha_{2n}^L, \alpha_{m1}^L, \cdots, \alpha_{mn}^L\right]^T$$

(12)

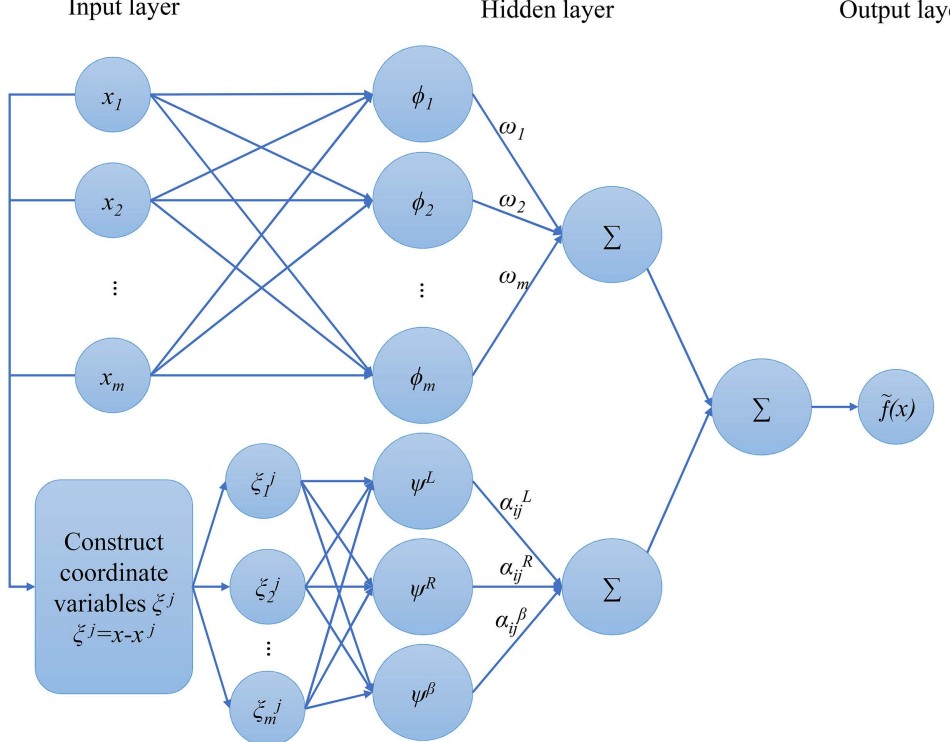

Input layer Hidden layer Output layer

**Fig 5. The E-RBF diagram.**

$$\alpha^R = \left[\alpha^R_{11}, \alpha^R_{12}, \cdots, \alpha^R_{1n}, \alpha^R_{21}, \cdots, \alpha^R_{2n}, \alpha^R_{m1}, \cdots, \alpha^R_{mn}\right]^T \qquad (13)$$

$$\alpha^\beta = \left[\alpha^\beta_{11}, \alpha^\beta_{12}, \cdots, \alpha^\beta_{1n}, \alpha^\beta_{21}, \cdots, \alpha^\beta_{2n}, \alpha^\beta_{m1}, \cdots, \alpha^\beta_{mn}\right]^T \qquad (14)$$

where the vectors $\alpha^L$, $\alpha^R$ and $\alpha^\beta$ are the matrix of $m \times n$ rows and 1 column.

Similar to the traditional RBF (See Equation (2)), to calculate the coefficients of nonradial basis functions, Equation (11) should satisfy the interpolation of the sampling points.

$$\sum_{i=1}^{m}\omega_i\phi_i\left(\|\boldsymbol{x}_k - \boldsymbol{x}_i\|\right)+\sum_{i=1}^{m}\sum_{j=1}^{n}\left[\alpha^L_{ij}\cdot\psi^L\left(x^j_k - x^j_i\right) + \alpha^R_{ij}\cdot\psi^R\left(x^j_k - x^j_i\right)\right.$$
$$\left.+\alpha^\beta_{ij}\cdot\psi^\beta\left(x^j_k - x^j_i\right)\right] = y\left(\boldsymbol{x}_k\right),\ \ k = 1, 2, \cdots, m \qquad (15)$$

In matrix notation, Equation (15) can be written as:

$$\left[\boldsymbol{\Phi}\right]\cdot\{\boldsymbol{\omega}\} + \left[\boldsymbol{\Psi}\right]\cdot\{\boldsymbol{\alpha}\} = \{\boldsymbol{Y}\} \qquad (16)$$

where $\left[\boldsymbol{\Phi}\right]$, $\{\boldsymbol{\omega}\}$ and $\{\boldsymbol{Y}\}$ are defined in Equation (3). And the matrix $\left[\boldsymbol{\Psi}\right]$ and $\{\boldsymbol{\alpha}\}$ can be described as:

$$\left[\mathbf{\Psi}\right] = \left[\mathbf{\Psi}^L, \mathbf{\Psi}^R, \mathbf{\Psi}^\phi\right], \{\alpha\} = \left[\left(\alpha^L\right)^T, \left(\alpha^R\right)^T, \left(\alpha^\beta\right)^T\right]^T \tag{17}$$

where $\alpha^L$, $\alpha^R$ and $\alpha^\beta$ can be obtained in Equations (12) to (14). $\mathbf{\Psi}^L$, $\mathbf{\Psi}^R$ and $\mathbf{\Psi}^\beta$ can be represented as:

$$\mathbf{\Psi}^L = \begin{bmatrix} \psi_{11}^L\left(x_1^1-x_1^1\right) & \psi_{12}^L\left(x_1^2-x_1^2\right) & \cdots & \psi_{1n}^L\left(x_1^n-x_1^n\right) & \psi_{2n}^L\left(x_1^n-x_2^n\right) & \cdots & \psi_{mn}^L\left(x_1^n-x_m^n\right) \\ \psi_{11}^L\left(x_2^1-x_1^1\right) & \psi_{12}^L\left(x_2^2-x_1^2\right) & \cdots & \psi_{1n}^L\left(x_2^n-x_1^n\right) & \psi_{2n}^L\left(x_2^n-x_1^n\right) & \cdots & \psi_{mn}^L\left(x_2^n-x_1^n\right) \\ \vdots & \vdots & \ddots & \vdots & \vdots & \ddots & \vdots \\ \psi_{m1}^L\left(x_m^1-x_1^1\right) & \psi_{m2}^L\left(x_m^2-x_1^2\right) & \cdots & \psi_{1n}^L\left(x_m^n-x_1^n\right) & \psi_{2n}^L\left(x_m^n-x_1^n\right) & \cdots & \psi_{mn}^L\left(x_m^n-x_1^n\right) \end{bmatrix}$$

$$\mathbf{\Psi}^R = \begin{bmatrix} \psi_{11}^R\left(x_1^1-x_1^1\right) & \psi_{12}^R\left(x_1^2-x_1^2\right) & \cdots & \psi_{1n}^R\left(x_1^n-x_1^n\right) & \psi_{2n}^R\left(x_1^n-x_1^n\right) & \cdots & \psi_{mn}^R\left(x_1^n-x_m^n\right) \\ \psi_{11}^R\left(x_2^1-x_1^1\right) & \psi_{12}^R\left(x_2^2-x_1^2\right) & \cdots & \psi_{1n}^R\left(x_2^n-x_1^n\right) & \psi_{2n}^R\left(x_2^n-x_1^n\right) & \cdots & \psi_{mn}^R\left(x_2^n-x_1^n\right) \\ \vdots & \vdots & \ddots & \vdots & \vdots & \ddots & \vdots \\ \psi_{m1}^R\left(x_m^1-x_1^1\right) & \psi_{m2}^R\left(x_m^2-x_1^2\right) & \cdots & \psi_{1n}^R\left(x_m^n-x_1^n\right) & \psi_{2n}^R\left(x_m^n-x_1^n\right) & \cdots & \psi_{mn}^R\left(x_m^n-x_1^n\right) \end{bmatrix}$$

$$\mathbf{\Psi}^\beta = \begin{bmatrix} \psi_{11}^\beta\left(x_1^1-x_1^1\right) & \psi_{12}^\beta\left(x_1^2-x_1^2\right) & \cdots & \psi_{1n}^\beta\left(x_1^n-x_1^n\right) & \psi_{2n}^\beta\left(x_1^n-x_2^n\right) & \cdots & \psi_{mn}^\beta\left(x_1^n-x_m^n\right) \\ \psi_{11}^\beta\left(x_2^1-x_1^1\right) & \psi_{12}^\beta\left(x_2^2-x_1^2\right) & \cdots & \psi_{1n}^\beta\left(x_2^n-x_1^n\right) & \psi_{2n}^\beta\left(x_2^n-x_1^n\right) & \cdots & \psi_{mn}^\beta\left(x_2^n-x_1^n\right) \\ \vdots & \vdots & \ddots & \vdots & \vdots & \ddots & \vdots \\ \psi_{m1}^\beta\left(x_m^1-x_1^1\right) & \psi_{m2}^\beta\left(x_m^2-x_1^2\right) & \cdots & \psi_{1n}^\beta\left(x_m^n-x_1^n\right) & \psi_{2n}^\beta\left(x_m^n-x_1^n\right) & \cdots & \psi_{mn}^\beta\left(x_m^n-x_1^n\right) \end{bmatrix}$$

Then, to simplify the formula, Equation (16) can be rewritten as:

$$\left[\mathbf{A}\right] \cdot \{\mathbf{\Theta}\} = \{\mathbf{Y}\} \tag{18}$$

where $\mathbf{A} = \left[\mathbf{\Phi}, \mathbf{\Psi}\right]$ and $\Theta = \left[\omega^T, \alpha^T\right]^T$. To obtain the Equation (18), the size of the matrix $\mathbf{A}$ is $m \times \left[m\left(3n+1\right)\right]$, and the size of the vector $\Theta$ is $m\left(3n+1\right) \times 1$.

## Solving approach for E-RBF equation

As mentioned above, the matrix $\mathbf{A}$ contains more columns than rows. That means that the Equation (18) is the underdetermined system equations, and it is difficult to solve these underdetermined system equations. Therefore, a pseudo-inverse approach is introduced to solve these underdetermined system of equations:

$$\{\mathbf{\Theta}\} = \left[\mathbf{A}\right]^+ \{\mathbf{Y}\} \tag{19}$$

where $\left[\mathbf{A}\right]^+$ denotes the pseudo-inverse matrix of $\left[\mathbf{A}\right]$. This pseudo-inverse approach yields a solution of $\Theta$, where $\Theta$ is the minimum norm solution. More expressions can be seen in literature [64]. Note that this is an interpolating hypersurface using the solved coefficients $\Theta$. After evaluating the coefficients $\Theta$, we can obtain the E-RBF based on Equation (11).

## Numerical examples and discussions

There are six test functions were used which were listed in Equation (20).

$$\begin{cases} f_1 = x_1 \sin x_1 + x_2 \sin x_2, \ -2\pi \le x_1, x_2 \le 2\pi \\ f_2 = (30 + x_1 \sin x_1)\left[4 + \exp\left(-x_2^2\right)\right], \\ 0 \le x_1 \le 10, \qquad 0 \le x_2 \le 6, \\ f_3 = x_1\sqrt{x_3^2 + 3} + x_2\sqrt{4x_3^2 + 1} + 0.05\ \exp x_3, \\ f_4 = x_1^2 \sin\left(x_2 + 2\right) + 10\cos x_3 + 4x_1 x_3 + 2x_2^2 + 3x_2, \\ f_5 = 0.1x_1^3 + 5x_3 - x_1\sin x_3 + 0.1\ \exp\left(x_2\right), \\ f_6 = 0.1x_2\ \exp\left(-x_1 + 2x_3\right) - 10x_3^2, \\ 0 \le x_1, x_2, x_3 \le 5 \end{cases} \quad (20)$$

To test the accuracy of the proposed E-RBF model with the adjusted widths for the basis functions, the standard error measure, the Root Mean Squared Error (RMSE) was utilized in this paper. The smaller the RMSE is, the better the accuracy of this approximation is. The value $RMSE = 0$ indicates perfect approximation.

$$RMSE = \left[\frac{1}{m}\sum_{i=1}^{m}\left[\tilde{f}\left(\boldsymbol{x}_i\right) - y\left(\boldsymbol{x}_i\right)\right]^2\right]^{\frac{1}{2}} \quad (21)$$

where $\tilde{f}\left(\boldsymbol{x}_i\right)$ illustrates the approximated value of $\boldsymbol{x}_i$ from RBF methods, $y\left(\boldsymbol{x}_i\right)$ indicates the actual value of $\boldsymbol{x}_i$ from the test functions. $m$ is the number of the sampling data.

In this paper, different sampling points are selected using Latin Hypercube Design, and 100 random test points are chosen to verify the accuracy of these numerical functions. Tables 3 and 4 displays the accuracy results under different sampling points for traditional RBF and E-RBF model. In Table 3, the width of the Gaussian basis function is 1.5.

From Tables 3 and 4, we can see that the E-RBF has better accurate results under different sampling points. And the more the sampling points are, the better the accuracy of the approximation model is. But the more the sampling points are, the larger the computation cost is.

Table 5 displays the accuracy results under different widths. From this table, it is obvious that different widths can affect the accuracy for RBF approximation, but the impact is not large. Furthermore, the RMSE for E-RBF with adjusted width $c$ is the smallest. That means this E-RBF with adjusted width has better accurate approximation.

## Adaptive E-RBF based interval analysis

### Traditional interval analysis

In the traditional interval analysis, we consider each parameter as interval value $a$ which describes the lower and upper bounds with respect to the perturbation of the uncertain factors.

**Table 3. The accuracy results under different sampling points for RBF.**

| Function No. | RMSE for RBF | | | |
| --- | --- | --- | --- | --- |
| | 100 sampling points | 200 sampling points | 500 sampling points | 1000 sampling points |
| $f_1$ | 0.0015 | 0.00148 | 0.00077 | 0.00003 |
| $f_2$ | 1.2672 | 0.9164 | 0.5824 | 0.4059 |
| $f_3$ | 0.2735 | 0.1737 | 0.1349 | 0.0977 |
| $f_4$ | 0.5378 | 0.3397 | 0.2590 | 0.1851 |
| $f_5$ | 0.2211 | 0.1393 | 0.1044 | 0.0789 |
| $f_6$ | 1.3069 | 1.0983 | 0.8135 | 0.6988 |

**Table 4. The accuracy results under different sampling points for E-RBF.**

| Function No. | RMSE for E-RBF | | | |
|---|---|---|---|---|
| | **100 sampling points** | **200 sampling points** | **500 sampling points** | **1000 sampling points** |
| $f_1$ | 0.0011 | 0.00104 | 0.00074 | 0.00002 |
| $f_2$ | 1.2663 | 0.9156 | 0.5809 | 0.4002 |
| $f_3$ | 0.2729 | 0.1727 | 0.1339 | 0.0918 |
| $f_4$ | 0.5024 | 0.3303 | 0.2458 | 0.1826 |
| $f_5$ | 0.2103 | 0.1380 | 0.0914 | 0.0781 |
| $f_6$ | 1.1633 | 1.0702 | 0.7484 | 0.6852 |

**Table 5. The accuracy results under different widths.**

| Function No. | RMSE for RBF | | | | RMSE for E-RBF |
|---|---|---|---|---|---|
| | **Permanence width c=0.5** | **Permanence width c=1.0** | **Permanence width c=1.5** | **Permanence width c=2.0** | **Adjusted width c** |
| $f_1$ | 0.0299 | 0.0177 | 0.0015 | 0.0013 | 0.0011 |
| $f_2$ | 1.2688 | 1.2850 | 1.2672 | 1.2749 | 1.2563 |
| $f_3$ | 0.2846 | 0.2735 | 0.2735 | 0.2826 | 0.2719 |
| $f_4$ | 0.5507 | 0.5437 | 0.5378 | 0.5206 | 0.5024 |
| $f_5$ | 0.2251 | 0.2219 | 0.2211 | 0.2246 | 0.2103 |
| $f_6$ | 1.4574 | 1.4236 | 1.3069 | 1.4812 | 1.1633 |

$$a^I = \left[a^L, a^U\right] = \left\{a \mid a^L \leq a \leq a^U, a \in R\right\} \tag{22}$$

where $a^L$ and $a^U$ indicate the lower and upper bounds, respectively. And the traditional interval analysis can be expressed below under two interval parameter $a^I$ and $b^I$.

$$\begin{aligned}
a^I + b^I &= \left[a^L + b^L, a^U + b^U\right] \\
a^I - b^I &= \left[a^L - b^U, a^U + b^L\right] \\
a^I \times b^I &= \left[\min\left(a^L \cdot b^L, a^L \cdot b^U, a^U \cdot b^L, a^U \cdot b^U\right),\right. \\
&\qquad \left. \max\left(a^L \cdot b^L, a^L \cdot b^U, a^U \cdot b^L, a^U \cdot b^U\right)\right] \\
a^I/b^I &= \left[a^L, a^U\right] \times \left[1/b^U, 1/b^L\right]
\end{aligned} \tag{23}$$

In matrix format, the interval vector of $n$-dimension can be defined as:

$$\boldsymbol{a}^I = \left[a_1^I, a_2^I, \cdots, a_n^I\right] \tag{24}$$

where $a_i^I$ represents $i$th interval value. And the general interval analysis can be described as:

$$\boldsymbol{y}^I = f\left(\boldsymbol{a}^I\right) \tag{25}$$

If the interval analysis contains both interval parameters and the ordinary real value, the general interval analysis should be written as:

$$\boldsymbol{y}^I = f^I\left(\boldsymbol{x}, \boldsymbol{a}^I\right) \tag{26}$$

where $\boldsymbol{y}^I$ is the interval vector of interval analysis results; $\boldsymbol{x}$ is the vector of the ordinary real parameters.

However, this traditional interval analysis will lead to overestimation problem especially for multiple terms of numerous interval variables [25,26,74]. In order to obtain more accurate bounds for the interval analysis, a nested-loop or double-loop optimization-based approach was introduced as follow.

$$
\begin{cases}
\boldsymbol{y}^I = \left[ \boldsymbol{y}^L, \boldsymbol{y}^U \right], \\
\boldsymbol{y}^L = \min_{\alpha \in \Gamma} f \left( \boldsymbol{x}, \boldsymbol{a}^I \right), \\
\boldsymbol{y}^U = \max_{\alpha \in \Gamma} f \left( \boldsymbol{x}, \boldsymbol{a}^I \right), \\
\Gamma = \left\{ \left( \boldsymbol{x}, \boldsymbol{a}^I \right) \middle| \boldsymbol{a}^L \leq \boldsymbol{a}^I \leq \boldsymbol{a}^U, \boldsymbol{x} \in R^n \right\}
\end{cases}
\tag{27}
$$

where $\boldsymbol{y}^L$ and $\boldsymbol{y}^U$ are vectors of lower and upper bounds under interval analysis.

## Critical points from E-RBF

Although the nested-loop or double-loop optimization-based approach can obtain precise bounds for the interval analysis, the number of optimization calculations will increase by $n$ times. As known for some complex structural problems, the accuracy of the approximate model owes to the approximation approach and the sampling method [63]. As mentioned above, the E-RBF is an effective approximate approach. This paper proposes another interval analysis method by critical points using E-RBF.

In mathematics, the largest or smallest points can be attained within the neighborhood of extrema points. Fortunately, the RBF is twice continuously differentiable function when employing Gaussian basis function for all $c \neq 0$. After the sampling points for the engineering problems are obtained, the analytic gradients and the Hessian matrix of second partial derivatives can calculated immediately.

Considering the E-RBF in Equation (11), the gradients of the equation are:

$$
\begin{aligned}
\frac{\partial \hat{f}(\boldsymbol{x})}{\partial \boldsymbol{x}} &= \sum_{i=1}^{m} \frac{\partial \omega_i \phi_i \left( \| \boldsymbol{x} - \boldsymbol{x}_i \| \right)}{\partial \boldsymbol{x}} + \sum_{i=1}^{m} \frac{\partial \psi_i \left( \boldsymbol{x} - \boldsymbol{x}_i \right)}{\partial \boldsymbol{x}} \\
&= \sum_{i=1}^{m} \frac{\partial \omega_i \phi_i \left( d \right)}{\partial d} \cdot \frac{\partial d}{\partial \boldsymbol{x}} + \sum_{i=1}^{m} \frac{\partial \psi_i \left( \xi_i \right)}{\partial \boldsymbol{x}}
\end{aligned}
\tag{28}
$$

where $\frac{\partial \phi_i(d)}{\partial d}$ and $\frac{\partial d}{\partial \boldsymbol{x}}$ can be expressed respectively below for the Gaussian basis function.

$$
\frac{\partial \phi_i(d)}{\partial d} = \left( -\frac{2d}{c^2} \right) \cdot \exp \left( -\frac{d^2}{c^2} \right)
$$

$$
\frac{\partial d}{\partial \boldsymbol{x}} = \frac{\sum_{j=1}^{n} \left( x^j - x_i^j \right)}{d}
\tag{29}
$$

where $x^j$ and $x_i^j$ represent the $j$th value of the design vector and the $i$th sampling vector, respectively. $n$ is the column number of the design vector. Then, the results of Equation (28) can be expressed as:

$$
\frac{\partial \hat{f}(\boldsymbol{x})}{\partial \boldsymbol{x}} = \left( -\frac{2}{c^2} \right) \cdot \exp \left( -\frac{d^2}{c^2} \right) \cdot \sum_{j=1}^{n} \left( x^j - x_i^j \right) + \sum_{i=1}^{m} \frac{\partial \psi_i \left( \xi_i \right)}{\partial \boldsymbol{x}}
\tag{30}
$$

where $\frac{\partial \psi_i(\xi_i)}{\partial \boldsymbol{x}} = \psi_i' \left( \xi_i \right) = \left( \psi^L \left( \xi_i \right) \right)' + \left( \psi^R \left( \xi_i \right) \right)' + \left( \psi^\beta \left( \xi_i \right) \right)'$ can be obtained from Table 6.

**Table 6. The derivative of nonradial basis function.**

| Region | Range of $\xi_i^j$ | $(\psi^L)'$ | $(\psi^R)'$ | $(\psi^\beta)'$ |
|--------|--------------------|-------------|-------------|-----------------|
| I | $\xi_i^j \leq -\gamma$ | $-n \cdot \gamma^{n-}$ | 0 | 1 |
| II | $-\gamma \leq \xi_i^j \leq 0$ | $n(\xi_i^j)^{n-1}$ | 0 | 1 |
| III | $0 \leq \xi_i^j \leq \gamma$ | 0 | $n(\xi_i^j)^{n-1}$ | 1 |
| IV | $\xi_i^j \geq \gamma$ | 0 | $n \cdot \gamma^{n-1}$ | 1 |

To obtain critical points from E-RBF, let Equation (29) be equal to zero.

$$\frac{\partial \hat{f}(\boldsymbol{x})}{\partial \boldsymbol{x}} = \left(-\frac{2}{c^2}\right) \cdot \exp\left(-\frac{d^2}{c^2}\right) \cdot \sum_{j=1}^{n}\left(x^j - x_i^j\right) + \sum_{i=1}^{m} \frac{\partial \psi_i(\xi_i)}{\partial \boldsymbol{x}} = 0 \tag{31}$$

After solve Equation (31), the critical point $\boldsymbol{x}_c$ ($\boldsymbol{x}_c = (x_{c1}, x_{c2}, \cdots, x_{cn})$) can be attained.

## Interval analysis by critical points from E-RBF

Similar to Equation (30), the Hessian matrix is the second partial derivatives that can be displays as:

$$H(\boldsymbol{x}) = \frac{\partial^2 \hat{f}(\boldsymbol{x})}{\partial \boldsymbol{x}^2} = \left(\frac{4n}{c^4}\right) \cdot \exp\left(-\frac{d^2}{c^2}\right) \cdot \sum_{j=1}^{n}\left(x^j - x_i^j\right) + \sum_{i=1}^{m} \frac{\partial^2 \psi_i(\xi_i)}{\partial \boldsymbol{x}^2} \tag{32}$$

where $n$ is the column number of the design vector; is the second derivative of nonradial basis function, which can be represented in Table 7.

When the critical point $\boldsymbol{x}_c$ is solved, put this critical point into Hessian matrix and judge the definition of the matrix $H(\boldsymbol{x}_c)$. The critical point is maximum or minimum if the Hessian matrix is definite positive or definite negative. Once the maximum and the minimum critical points are attained, the lower and upper bounds for the interval analysis is confirmed.

## Adaptive E-RBF based interval analysis

The detailed algorithm for the adaptive E-RBF based interval analysis method is described in Fig 6. Due to the satisfaction of the engineering accuracy, the stopping criterion can be illustrated as:

$$\begin{cases} \left|\frac{f^L(k) - f^L(k-1)}{f^L(k-1)}\right| \leq 0.01 \\ \left|\frac{f^U(k) - f^U(k-1)}{f^U(k-1)}\right| \leq 0.01 \end{cases} \tag{33}$$

From Fig 6, the process of adaptive E-RBF based interval analysis can be listed below:

**Table 7. The second derivative of nonradial basis function.**

| Region | Range of $\xi_i^j$ | $(\psi^L)''$ | $(\psi^R)''$ | $(\psi^\beta)''$ |
|--------|--------------------|--------------|--------------|------------------|
| I | $\xi_i^j \leq -\gamma$ | 0 | 0 | 0 |
| II | $-\gamma \leq \xi_i^j \leq 0$ | $n(n-1)(\xi_i^j)^{n-2}$ | 0 | 0 |
| III | $0 \leq \xi_i^j \leq \gamma$ | 0 | $n(n-1)(\xi_i^j)^{n-2}$ | 0 |
| IV | $\xi_i^j \geq \gamma$ | 0 | 0 | 0 |

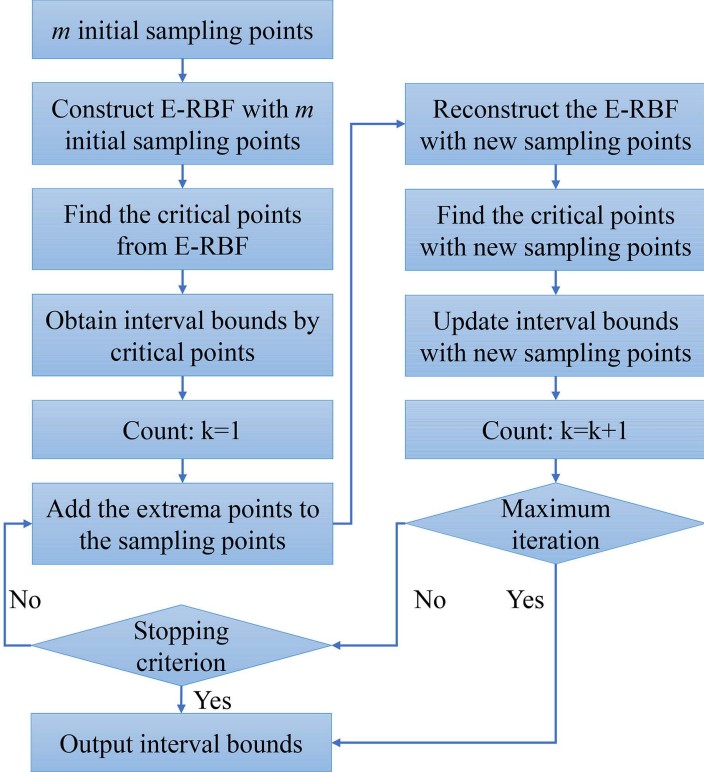

**Fig 6. The adaptive E-RBF based interval analysis.**

Step 1: Attain $m$ initial sampling points and sampling response from engineering problems, and construct E-RBF model with these $m$ initial sampling points.

Step 2: Find the critical points from E-RBF by Equation (30), and obtain interval bounds from critical points based on Equation (31).

Step 3: Count k = 1, then add these extrema points to the sampling points. Update the sampling points and reconstruct the E-RBF model with new sampling points.

Step 4: Solve the critical points using new sampling points. Update interval bounds from the critical points, and count k=k+1.

Step 5: Test whether the maximum iteration is satisfied. If the maximum iteration is satisfied, output the interval bounds. If not, test the stopping criterion is satisfied.

Step 6: If the stopping criterion is satisfied, output the interval bounds. If not, return to step 5 until the maximum iteration or stopping criterion are satisfied.

In this process of adaptive E-RBF based interval analysis, the maximum iteration can be chosen as 10–20 to get accurate interval bounds.

## Comparison to contemporary work

### Comparison 1: Test function

The first numerical test problem is given as:

$$\begin{cases} f(x, a) = -24 \sin\left(\sqrt{\left(\frac{x-40}{2}\right)^2 + (a-5)^2}\right) / \sqrt{\left(\frac{x-40}{2}\right)^2 + (a-5)^2} + 46 \\ 24 \leq x \leq 43, \ a \in a^I = [3.5, 16.5] \end{cases} \tag{34}$$

This problem was also performed as a numerical test by Z. Zhao [61] using RBF model and local-densifying method. In this paper, the size of the sampling points can be chosen as 20 for problem 1. And the Latin hypercube design (LHD) was carried out here to get the initial sampling points. Then the sampling points should be normalized into 0–1 range to avoid the influence of different magnitudes of variables. The maximum iteration can be recommended as 10.

Table 8 shows the interval analysis results between accurate function, Zhao's method and the adaptive approach based interval analysis using adaptive E-RBF and RBF under $x = 24$, $40.03$ and $42.89$. Fig 7 illustrates the comparison responses of the test function, the adaptive method in RBF and the adaptive method in this paper under the uncertain parameter range of $a \in a^I = [3.5, 16.5]$. Among all these methods, the method proposed in this paper is almost similar to the shape of the original function in Fig 7. That is to say, compared with Zhao's method and the method in RBF, the method in this paper has more accurate and reliable bounds of uncertainty in the uncertain interval analysis.

In Table 9, $a^L$ and $a^U$ are the solutions of the uncertain interval parameter $a$ for the minimum and maximum objective bounds under $x$, respectively. NI in Table 9 is the number of the iteration for the adaptive interval analysis calculation. From these two tables, it can be seen that the bounds of the interval analysis in this paper are very similar to the bounds in accurate function. And the number of the iteration in this paper is much lesser than in Zhao's method. This indicates that the efficiency of this adaptive E-RBF interval analysis method is improved on the basis of no deterioration in accuracy.

## Comparison 2: Belleville spring

The second example is a Belleville spring problem that is also performed by L. Wang [75] and is shown in Fig 8.

In this problem, the external diameter $d_e$, the internal diameter $d_i$, the thickness $t$ and the free height $h$ are considered as four interval uncertain parameters. The performance of this Belleville spring is the rated load $F$ (N) and the maximum stress $\sigma_{max}$ (Pa), which can be formulated below:

$$\begin{aligned} F &= \frac{E\delta_{max}}{(1-\nu^2)\alpha(d_e/2)^2}\left[\left(h - \frac{\delta_{max}}{2}\right)(h - \delta_{max})t + t^3\right] \\ \sigma_{max} &= \frac{E\delta_{max}}{(1-\nu^2)\alpha(d_e/2)^2}\left[\beta\left(h - \frac{\delta_{max}}{2}\right) + \gamma \cdot t\right] \end{aligned} \tag{35}$$

where $E$ is Young's modulus; $\delta_{max}$ is the maximum allowable deflection; $\nu$ is Poisson's ratio. In this example, $E$, $\delta_{max}$ and $\nu$ can be chosen as 210 GPa, $h$ and 0.3, respectively [75]. And the coefficients in these functions can be expressed as:

$$\alpha = \frac{6}{\pi \ln K}\left(\frac{K-1}{K}\right)^2, \ \beta = \frac{6}{\pi \ln K}\left(\frac{K-1}{\ln K} - 1\right), \ \gamma = \frac{6}{\pi \ln K}\left(\frac{K-1}{2}\right) \tag{36}$$

**Table 8. The interval analysis results between two methods.**

| Item | 1 | | 2 | | 3 | |
|---|---|---|---|---|---|---|
| | $f^L$ | $f^U$ | $f^L$ | $f^U$ | $f^L$ | $f^U$ |
| $x$ | 24 | | 40.03 | | 42.89 | |
| Accurate method | 43.03 | 48.19 | 22.01 | 51.21 | 29.65 | 51.19 |
| Zhao's method | 43.03 | 48.19 | 22.01 | 51.21 | 29.49 | 48.74 |
| Method using RBF | 43.09 | 48.17 | 22.37 | 50.67 | 29.55 | 51.93 |
| Method in this paper | 43.03 | 48.18 | 22.04 | 51.19 | 29.69 | 50.98 |

 

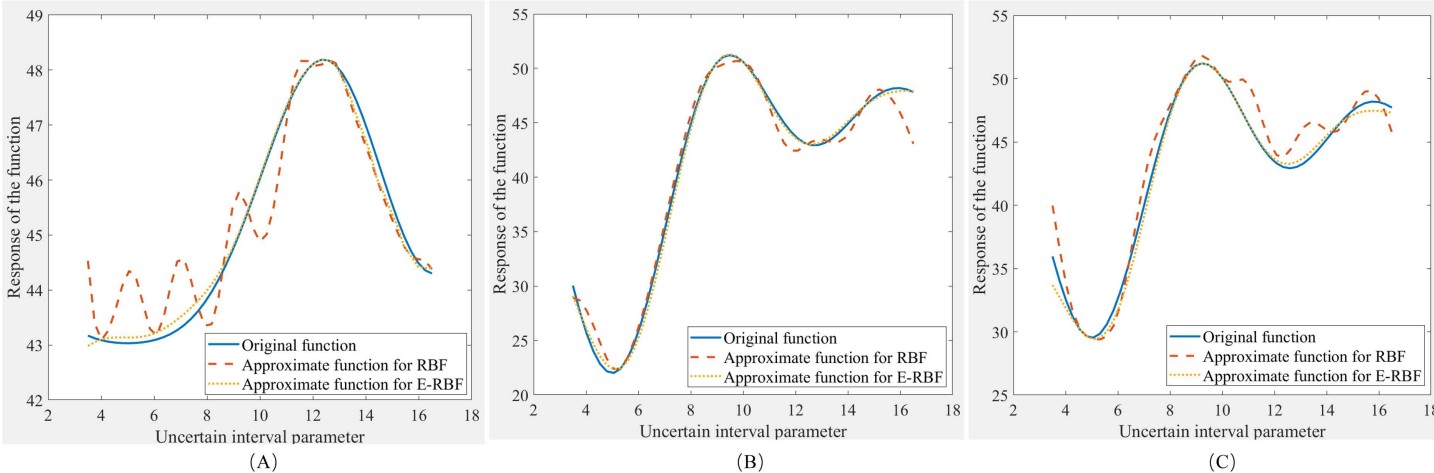

**Fig 7. The response under interval parameter.**

**Table 9. The results of uncertain parameter and the number of iteration.**

| Item | 1 | | 2 | | 3 | |
|---|---|---|---|---|---|---|
| | $a^L$ | $a^U$ | $a^L$ | $a^U$ | $a^L$ | $a^U$ |
| $x$ | 24 | | 40.03 | | 42.89 | |
| Accurate function | 5.06 | 12.34 | 5.06 | 9.48 | 4.80 | 9.35 |
| Zhao's method | 4.96 | 12.41 | 4.95 | 9.50 | 4.94 | 9.61 |
| NI for Zhao's method | 8 | | 8 | | 5 | |
| Method in RBF | 4.02 | 12.60 | 5.10 | 9.74 | 4.78 | 9.61 |
| NI for Method in RBF | 5 | | 2 | | 2 | |
| Method in this paper | 4.96 | 12.04 | 5.06 | 9.58 | 4.81 | 9.28 |
| NI for Method in this paper | 4 | | 2 | | 3 | |

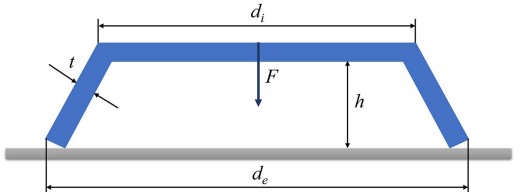

**Fig 8. The diagram of Belleville spring.**

$$K = \frac{d_e}{d_i} \tag{37}$$

In the review [64], L. Wang proposed a novel feedforward neural network method named FANNBIA. With this method, first-order (FO) and second-order (SO) Taylor model was utilized to compare the accuracy and the efficiency for structural solution bounds. Furthermore, the nominal uncertain parameters of $d_e$, $d_i$, $t$, and $h$ are 0.3 m, 0.211 m, 7.272 mm,

and 5.0 mm, respectively. The uncertain level for uncertain interval parameters can be selected as 10%, that means the interval parameters are [0.27, 0.33], [0.1899, 0.2321], [6.5448, 7.9992], and [4.5, 5.5]. Table 10 displays the comparison between FANNBIA models and this paper's method.

From Table 10, the method in this paper has relative accurate interval analysis solution bounds compared to the accurate value and the FANNBIA. This implies this paper's method has more efficiency than other interval analysis methods.

## Comparison 3: A cantilever tube

In this problem, a cantilever tube is subjected to three external forces $F_1$, $F_2$, $P$, and a torsion $T$ (See Fig 9). Thus, the maximum von Mises stress can be treated as the performance result, which can be stated as:

$$\sigma_{vms} = \sqrt{\sigma_x^2 + 3\tau_{zx}^2} \tag{38}$$

where $\sigma_x$ and $\tau_{zx}$ are the normal stress and the torsional stress, respectively. Then the normal stress can be calculated below.

$$\sigma_x = \frac{P + F_1 \sin\theta_1 + F_2 \sin\theta_2}{A} + \frac{M \cdot d}{2I} \tag{39}$$

where $M$ is the bending moment, which can be expressed by:

$$M = F_1 L_1 \cos\theta_1 + F_2 L_2 \cos\theta_2 \tag{40}$$

**Table 10. The interval analysis comparison results.**

| Item | Rated load $F$ | | Maximum stress $\sigma_{max}$ | |
|---|---|---|---|---|
| | $F^L$ in 10% range (KN) | $F^U$ in 10% range (KN) | $\sigma_{max}{}^L$ in 10% range (GPa) | $\sigma_{max}{}^U$ in 10% range (GPa) |
| Accurate value | 17.17 | 143.33 | 0.6705 | 3.0823 |
| FANNBIA (FO) | 17.11 | 140.63 | 0.6693 | 3.035 |
| FANNBIA (SO) | 17.03 | 142.92 | 0.668 | 3.0756 |
| Method in this paper | 17.07 | 141.62 | 0.6681 | 2.997 |

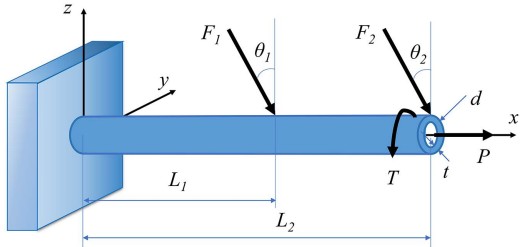

**Fig 9. The diagram of cantilever tube.**

*A* and *I* are the area and the inertia moment, respectively.

$$A = \frac{\pi}{4}\left[d^2 - (d - 2t)^2\right], \quad I = \frac{\pi}{64}\left[d^4 - (d - 2t)^4\right]$$

(41)

The torsional stress can be listed as:

$$\tau_{zx} = \frac{T \cdot d}{4I}$$

(42)

L. Wang [75] modified this problem by adding five interval uncertain parameters $t$, $F_1$, $F_2$, $P$ and $T$. $d$ and $t$ are the thickness of the diameter and the cantilever tube, respectively. $\theta_1$ and $\theta_2$ are the angles of the external forces $F_1$ and $F_2$, respectively. $L_1$ and $L_2$ are the distances of the external forces $F_1$ and $F_2$, respectively. These parameters for this problem can be defined as nominal values, which $d$, $\theta_1$, $\theta_2$, $L_1$ and $L_2$ are 42 mm, 5°, 10°, 60 mm and 120 mm. And two levels (10% and 20%) of the uncertain interval parameters can be chosen to compare the different errors between accurate bounds and solution bounds from interval analysis.

Table 11 illustrates the accuracy of different interval analysis solution bounds methods. In this table, errors are the relative errors which express the accuracy of each interval analysis methods. From Table 11, even though the errors in this paper are not all minimal on the two bounds, the accuracy of this paper acceptable compared to FANNBIA methods. And this proposed method in the paper has some advantage in terms of efficiency. Again, the proposed analysis method in this paper is proved to be accurate and efficient.

## Comparison 4: A composite shell

This part puts forward a famous composite shell as a practical example to withstand the pressure under the water [60]. In this underwater hull, the buckling performance may be the most significant impact under the deep water pressure [76]. And the stacking sequence is one of the important terms for buckling performance [60].

In this example, G. Zheng [60] constructed a double-loop interval optimization by the progressive trigonometric mixed response surface method (PTMRSM) for the optimal design of the stacking sequence of the composite shell. Fig 10 is the diagram of this composite shell problem. In this problem, 10 symmetric stacking sequence is selected following G. Zheng, which means 5 different layer angles is chosen as the design variables.

In order to obtain these performance variables using ANSYS Workbench (See Fig 11). Under this ANSYS Workbench buckling simulation process, 5 layer angles ($\theta_1,\theta_2,\theta_3,\theta_4,\theta_5$) and 1 thickness ($t$) can be treated as the input variables, and the critical buckling pressure $P_{cr}$ can be considered as the buckling objective performance of this composite shell. Thus there are corresponding functional relationships for input variables and objective performance as below.

$$\begin{cases} P_{cr} = f\left(\overrightarrow{\theta}, t\right), \overrightarrow{\theta} = \left[\theta_1, \theta_2, \theta_3, \theta_4, \theta_5\right], \\ -90^o \leq \theta_i \leq 90^o, i = 1, 2, 3, 4, 5, \\ t \in t^I = \left[0.5, 0.7\right] mm \end{cases}$$

(43)

**Table 11. The accuracy of different methods' interval analysis solution bounds.**

| Item | $\sigma_{VMS}{}^L$ in 10% range (MPa) | $\sigma_{VMS}{}^R$ in 10% range (MPa) | $\sigma_{VMS}{}^L$ in 20% range (MPa) | $\sigma_{VMS}{}^R$ in 20% range (MPa) |
|---|---|---|---|---|
| Errors in this paper (%) | 0.2351 | 0.2353 | 1.203 | 1.009 |
| Errors in FANNBIA (FO) (%) | 0.2044 | 0.2491 | 0.7533 | 1.1241 |
| Errors in FANNBIA (SO) (%) | 0.4153 | 0.0794 | 1.5577 | 0.821 |

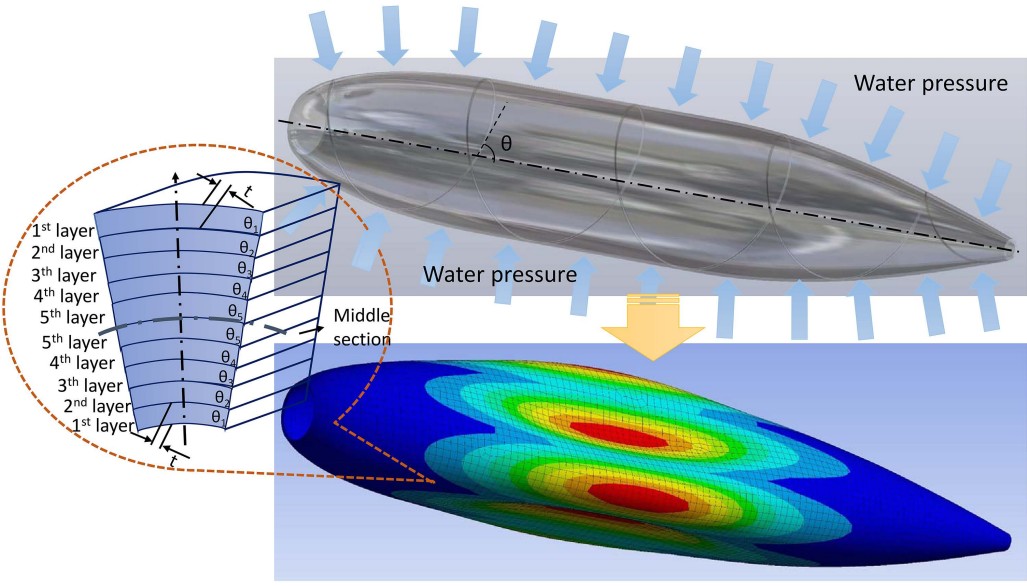

**Fig 10. The diagram of Composite shell.**

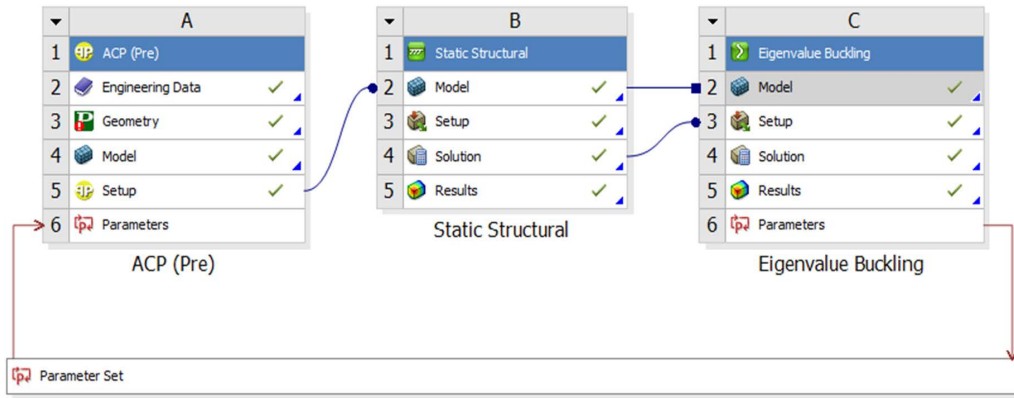

**Fig 11. The diagram of ANSYS Workbench for performance.**

For the buckling simulation process, 200 optimal LHD sample points was carried out, and 400 optimal LHD sample points was performed for the maximum deformation displacement simulation calculation. In review [60], $\vec{\theta}_1$ =[1.55, 14.74, 32.95, 86.56, −90], $\vec{\theta}_2$ = [−14.86, 2.48, 8.94, −4.84, −90], $\vec{\theta}_3$ = [−16.13, 1.31, 14.84, 16.68, −89.82], $\vec{\theta}_4$ = [−12.68, 0.56, 10.30, 21.71, −90] were the optimal layer angles for different preset possibility level. Therefore, there are a comparison between Zheng's double-loop interval analysis by PTMRSM, this paper's interval analysis method and Finite Element Method for short FEM (See Table 12).

From Table 12, NI is the number of the iteration for the interval analysis computation of upper and lower bounds, FEM represents the Finite Element Method, which comes from ANSYS Workbench. Compared to double-loop interval analysis with PTMRSM, though the two methods are different, this paper's method can obtain almost the same results of layer angles. Furthermore, the number of iteration for the interval computation in this paper is much lesser than Zheng's double-loop

**Table 12. The solution bounds of different interval analysis methods.**

| $\vec{\theta}$ | $f^L$ for Zheng | $f^U$ for Zheng | NI for Zheng | $f^L$ in this paper | $,f^U$ in this paper | NI in this paper | $f^L$ in FEM | $f^U$ in FEM |
|---|---|---|---|---|---|---|---|---|
| $\vec{\theta}_1$ | 1.2400E+06 | 2.1240E+06 | 10 | 1.1083E+06 | 2.1321E+06 | 3 | 1.1350E+06 | 2.1090E+06 |
| $\vec{\theta}_2$ | 1.0547E+06 | 1.7637E+06 | 10 | 1.0449E+06 | 1.6711E+06 | 2 | 9.2357E+05 | 1.6492E+06 |
| $\vec{\theta}_3$ | 9.7618E+05 | 1.6562E+06 | 10 | 9.0867E+05 | 1.5999E+06 | 2 | 9.0219E+05 | 1.5643E+06 |
| $\vec{\theta}_4$ | 9.5483E+05 | 1.6256E+06 | 10 | 8.7442E+05 | 1.5688E+06 | 2 | 8.0746E+05 | 1.5250E+06 |

interval analysis by PTMRSM. In contrast to double-loop interval analysis with PTMRSM, the bounds in this paper are more similar to the FEM analysis results which means the bounds in this paper have more accurate analysis results.

## Conclusions

This paper proposes an adaptive Extended Radial Basis Function (E-RBF) based interval analysis method to improve the accuracy and the efficiency of the calculation of upper and lower solution bounds under uncertain factors in engineering structures. In this uncertain interval analysis, the E-RBF method was performed with the help of adjusted widths of the Gaussian basis function, which can change the constant $c$ to an adjusted constant for basis functions based on the adaptive scaling technique and each distance between each sampling points. In order to obtain the accurate bounds for the interval analysis, the process of adaptive E-RBF based interval analysis is proposed using the critical points and the adaptive interval analysis method. Then numerical functions and comparison of engineering structures were utilized to test the accuracy and efficiency of this method. The principal conclusions of this paper are outlined below:

1. The RMSE for E-RBF with adjusted widths $c$ is the smallest. It implies the E-RBF with adjusted widths has better accurate approximation.

2. The accuracy of the E-RBF with adjusted widths of the Gaussian basis function for all numerical functions is much better than the traditional RBF model under different sampling points. Furthermore, different widths can affect the accuracy for RBF approximation, but the impact is not large.

3. From the comparison of Zhao's test function, the bounds of the interval analysis in this paper are very similar to the bounds in accurate function. And the number of the iteration in this paper is much lesser than in Zhao's method. It means that the efficiency of this adaptive E-RBF interval analysis method is improved on the basis of no deterioration in accuracy.

4. Among Zhao's method and the adaptive interval analysis method in RBF and E-RBF, the method proposed in this paper (adaptive interval analysis method in E-RBF) is almost similar to the shape of the original function. That is to say, compared with Zhao's method and the method in RBF, the method in this paper has more accurate and reliable bounds of uncertainty in the uncertain interval analysis.

5. From the comparison of the Belleville spring and the cantilever tube problem, the method in this paper has relative accurate interval analysis solution bounds compared to the accurate value and the FANNBIA. That is to say, this paper's method is more accurate than other interval analysis methods. Again, the proposed analysis method in this paper is proved to be accurate and efficient.

6. Compared to double-loop interval analysis with PTMRSM, though the two methods are different, this paper's method can obtain almost the same results of layer angles. Furthermore, the number of iteration for the interval computation in this paper is much lesser than Zheng's double-loop interval analysis by PTMRSM.

7. In contrast to double-loop interval analysis with PTMRSM, the bounds in this paper are more similar to the FEM analysis results which means the bounds in this paper have more accurate analysis results.

Furthermore, it has to be mentioned that this paper only focuses on the explanation of the adaptive E-RBF based interval analysis method. Only six numerical functions and three structural problems and one composite shell are performed to verify this proposed method. But how the E-RBF outperforms other surrogate models (e.g., Kriging or response surface methods) in specific contexts will be our next goal and future studies.

## Supporting information

**S1 File. Minimal Data Set.**
(MAT)

## Acknowledgments

The authors wish to thank Dr. G. Zheng for sharing the E-RBF Matlab code. The authors would also like to thank all the reviewers for their valuable suggestions.

## Author contributions

**Conceptualization:** Xu Jitang.

**Data curation:** Xu Jitang.

**Formal analysis:** He Jinli.

**Investigation:** He Jinli.

**Resources:** He Jinli.

**Software:** Chen Qiang.

**Supervision:** Chen Qiang.

**Validation:** Chen Qiang.

**Writing – original draft:** Xu Jitang.

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
