## [Decision Letter · Decision Letter 0]

8 Jul 2025

Dear Dr. Xu,

Thank you for submitting your manuscript to PLOS ONE. After careful consideration, we feel that it has merit but does not fully meet PLOS ONE’s publication criteria as it currently stands. Therefore, we invite you to submit a revised version of the manuscript that addresses the points raised during the review process.

**ACADEMIC EDITOR: Author requested to address the comments given by reviewers and submit the manuscript.**

We look forward to receiving your revised manuscript.

Kind regards,

ARUNKUMAR C, Ph.D

Academic Editor

PLOS ONE

Journal Requirements:

https://journals.plos.org/plosone/s/file?id=ba62/PLOSOne_formatting_sample_title_authors_affiliations.pdf....

[Beichuang Teaching Assistant Program 2021BCE02010].

Additional Editor Comments:

Author requested revised as per the suggestion given by the reviewer

**Comments from PLOS Editorial Office**: We note that one or more reviewers has recommended that you cite specific previously published works in the current and previous rounds of revision. As always, we recommend that you evaluate the requested works to determine whether they are relevant and should be cited. It is not a requirement to cite these works and you may remove any added citations before the manuscript proceeds to publication. We appreciate your attention to this request.: We note that one or more reviewers has recommended that you cite specific previously published works in the current and previous rounds of revision. As always, we recommend that you evaluate the requested works to determine whether they are relevant and should be cited. It is not a requirement to cite these works and you may remove any added citations before the manuscript proceeds to publication. We appreciate your attention to this request.: We note that one or more reviewers has recommended that you cite specific previously published works in the current and previous rounds of revision. As always, we recommend that you evaluate the requested works to determine whether they are relevant and should be cited. It is not a requirement to cite these works and you may remove any added citations before the manuscript proceeds to publication. We appreciate your attention to this request.: We note that one or more reviewers has recommended that you cite specific previously published works in the current and previous rounds of revision. As always, we recommend that you evaluate the requested works to determine whether they are relevant and should be cited. It is not a requirement to cite these works and you may remove any added citations before the manuscript proceeds to publication. We appreciate your attention to this request.

Reviewers' comments:

Reviewer's Responses to Questions

**Comments to the Author**

1. Is the manuscript technically sound, and do the data support the conclusions?

Reviewer #1: Yes

Reviewer #2: Yes

Reviewer #3: Yes

2. Has the statistical analysis been performed appropriately and rigorously?

Reviewer #1: Yes

Reviewer #2: Yes

Reviewer #3: Yes

3. Have the authors made all data underlying the findings in their manuscript fully available?

Reviewer #1: Yes

Reviewer #2: Yes

Reviewer #3: Yes

4. Is the manuscript presented in an intelligible fashion and written in standard English?

Reviewer #1: Yes

Reviewer #2: No

Reviewer #3: Yes

Reviewer #1: The main title is well developed. The main idea is well developed by chronological order. The main idea is following STEM very well. The idea was developed the structural engineering and engineering technology very well.

Reviewer #2: The authors are required to comprehensively address the following suggestions:

1. I recommend expanding the abstract to within 200–250 words, focusing on the core aim, methodology, key results, and conclusions.

2. Revise in the abstract (To test the accuracy and efficiency of the E-RBF with adjusted widths, six numerical functions are chosen.).

3. The manuscript contains numerous grammatical errors.

4. The paper addresses a significant challenge in structural engineering: handling uncertainties in interval analysis for complex systems. The introduction effectively highlights the importance of uncertainty quantification, distinguishing between aleatory and epistemic uncertainties, and justifies the need for non-probabilistic methods like interval analysis. However, the motivation for focusing specifically on the adaptive E-RBF method could be strengthened by explicitly stating how it addresses limitations of existing methods (e.g., computational inefficiency or inaccuracy in large interval ranges) earlier in the introduction https://doi.org/10.1016/j.jobe.2025.113302 - https://doi.org/10.1038/s41598-025-06725-z - https://doi.org/10.1016/j.jobe.2025.113302.

5. The novelty is well-demonstrated through comparisons with traditional RBF and other methods, though the paper could further clarify how the E-RBF outperforms other surrogate models (e.g., Kriging or response surface methods) in specific contexts.

6. The objective, methodology, and results should be better described, discussed and justified.

7. The manuscript contains excessive abbreviations. Only the necessary ones should be retained.

8. Equations (1) to (32) provide a clear derivation of the E-RBF and its application to interval analysis. However, the explanation of the pseudo-inverse approach for solving underdetermined systems (Equation 19) is brief and lacks discussion on numerical stability or potential issues (e.g., ill-conditioning of the matrix A).

9. Sorry revise the capture of all figures.

10. The paper lacks quantitative data on actual computational time or resource usage, which would provide a more concrete evaluation of efficiency. Including such metrics would strengthen the claims.

11. Revise Fig. 5.

12. Ensure the accurate and consistent use of superscripts and subscripts throughout the manuscript. Any improper formatting or irregularities may indicate that the text was generated or modified using AI tools.

13. Compare the results to contemporary work in a separate section, namely, comparison to contemporary work.

14. For the conclusion it is so short; please update it and add academic numbers and percentages.

15. Update all references to ensure they are current and relevant.

Reviewer #3: This paper presents a technically promising approach to interval analysis using an adaptive E-RBF method. However, major revisions are needed to improve clarity, correct language issues, and strengthen comparative validation before it can be considered for publication in a reputed journal like PLOS ONE.

1. A thorough language edit is required. Consider using professional editing services before resubmission.

2. Lack of Benchmark Comparisons. The method is mainly compared to Zhao’s work; broader comparisons to other interval methods, e.g., Kriging, fuzzy-set methods, would strengthen the paper. Please include additional benchmarks or at least a discussion on how the proposed method compares in broader contexts.

3. Ensure all figures and tables are included, clearly labelled, and cross-referenced correctly in the text.

4. The mathematics is extensive but lacks deeper theoretical validation (e.g., convergence proof, sensitivity analysis). Add theoretical justification or boundary conditions under which E-RBF guarantees convergence or uniqueness.

.

Reviewer #1: **Yes:** Norshida Abdul KadirNorshida Abdul KadirNorshida Abdul KadirNorshida Abdul Kadir

Reviewer #2: **Yes:** Mohamed AbdellatiefMohamed AbdellatiefMohamed AbdellatiefMohamed Abdellatief

Reviewer #3: **Yes:** Dr.R.RajkumarDr.R.RajkumarDr.R.RajkumarDr.R.Rajkumar

---

## [Author Response · Author response to Decision Letter 1]

25 Jul 2025

Comment Response

I recommend expanding the abstract to within 200–250 words, focusing on the core aim, methodology, key results, and conclusions. Thank you for your support. And I have revised the Abstract to 200-250 words. See Page 1 and 2, Line 16 to Line 32.

Revise in the abstract (To test the accuracy and efficiency of the E-RBF with adjusted widths, six numerical functions are chosen.). Thank you for your suggestion, we revised this sentence in the abstract.

See Page 1 Line 24 to 26.

The manuscript contains numerous grammatical errors. Thank you for your suggestion, we have comprehensively optimized the article writing and grammar issues.

The paper addresses a significant challenge in structural engineering: handling uncertainties in interval analysis for complex systems. The introduction effectively highlights the importance of uncertainty quantification, distinguishing between aleatory and epistemic uncertainties, and justifies the need for non-probabilistic methods like interval analysis. However, the motivation for focusing specifically on the adaptive E-RBF method could be strengthened by explicitly stating how it addresses limitations of existing methods (e.g., computational inefficiency or inaccuracy in large interval ranges) earlier in the introduction https://doi.org/10.1016/j.jobe.2025.113302 - https://doi.org/10.1038/s41598-025-06725-z Thank you for your suggestion. We added an expression about motivation in the Introduction.

See Page 4 to 5, Line 86 to 93.

"However, double-loop or nested-loop optimization-based process means more intensive computation. In order to save the computational cost, the approximation or surrogate model should be introduced for the nested-loop or double-loop interval analysis. M. Abdellatief [53] presented a machine-learning technique to address the limitations of the complex, non-linear interactions between different mixing design parameters. To balance accuracy and efficiency, there is artificial intelligence method to predict the early-age compressive strength for optimal mix design [54]. And these reviews are the motivation of the E-RBF method."

The novelty is well-demonstrated through comparisons with traditional RBF and other methods, though the paper could further clarify how the E-RBF outperforms other surrogate models (e.g., Kriging or response surface methods) in specific contexts. Thanks for the comment. This will be our next goal.

See Page 37 and 38, Line 628 to 630.

The objective, methodology, and results should be better described, discussed and justified. Thanks for the comment. I have revised the objective, methodology, and results.

The manuscript contains excessive abbreviations. Only the necessary ones should be retained. Thanks for the comment. I have deleted the excessive abbreviations.

Equations (1) to (32) provide a clear derivation of the E-RBF and its application to interval analysis. However, the explanation of the pseudo-inverse approach for solving underdetermined systems (Equation 19) is brief and lacks discussion on numerical stability or potential issues (e.g., ill-conditioning of the matrix A). Thank you for your comment.

We add an expression in Page 18, Line 312 to 313.

The paper lacks quantitative data on actual computational time or resource usage, which would provide a more concrete evaluation of efficiency. Including such metrics would strengthen the claims. Thank you for your comment.

We add the computational resources. See Table 9 in Page 27, Line 453 and Table 12 in Page 35, Line 581

Revise Fig. 5. Thank you for your comment.

We have revised Fig. 5 in Page 16, Line 280.

Ensure the accurate and consistent use of superscripts and subscripts throughout the manuscript. Any improper formatting or irregularities may indicate that the text was generated or modified using AI tools. Thank you for your comment.

We have ensured all the superscripts and subscripts.

Compare the results to contemporary work in a separate section, namely, comparison to contemporary work. Thank you for your comment.

We have separate this section, Page 26 to 35, Line 441 to 582.

For the conclusion it is so short; please update it and add academic numbers and percentages. Thank you for your comment.

We have revised the conclusion, Page 35 to 38, Line 584 to 624.

Update all references to ensure they are current and relevant. Thank you for your comment. We have updated the references.

Comment Response

A thorough language edit is required. Consider using professional editing services before resubmission. Thank you for your support. We have updated the thorough language.

Lack of Benchmark Comparisons. The method is mainly compared to Zhao’s work; broader comparisons to other interval methods, e.g., Kriging, fuzzy-set methods, would strengthen the paper. Please include additional benchmarks or at least a discussion on how the proposed method compares in broader contexts. Thank you for your suggestion. we have added a comparison between accurate function and adaptive interval analysis method in RBF.

See Page 27 and 28 Line 451 to 465.

Ensure all figures and tables are included, clearly labelled, and cross-referenced correctly in the text. Thank you for your suggestion, we have ensured all the figrures and tables.

The mathematics is extensive but lacks deeper theoretical validation (e.g., convergence proof, sensitivity analysis). Add theoretical justification or boundary conditions under which E-RBF guarantees convergence or uniqueness. Thank you for your suggestion. We have added a comparison between RBF and E-RBF, See Page 19 to 20 and 27 to 28.

---

## [Decision Letter · Decision Letter 1]

11 Mar 2026

An Adaptive Extended Radial Basis Function based Interval Analysis Method for Structural Engineering Solutions

PONE-D-25-23864R1

Dear Dr. Xu,

We’re pleased to inform you that your manuscript has been judged scientifically suitable for publication and will be formally accepted for publication once it meets all outstanding technical requirements.

Kind regards,

Kaywan Othman Ahmed

Academic Editor

PLOS One

Additional Editor Comments (optional):

Reviewers' comments:

Reviewer's Responses to Questions

**Comments to the Author**

Reviewer #2: All comments have been addressed

Reviewer #4: (No Response)

2. Is the manuscript technically sound, and do the data support the conclusions?

Reviewer #2: Yes

Reviewer #4: (No Response)

3. Has the statistical analysis been performed appropriately and rigorously?

Reviewer #2: Yes

Reviewer #4: (No Response)

4. Have the authors made all data underlying the findings in their manuscript fully available?

Reviewer #2: Yes

Reviewer #4: (No Response)

5. Is the manuscript presented in an intelligible fashion and written in standard English?

Reviewer #2: Yes

Reviewer #4: (No Response)

Reviewer #2: The authors have made commendable improvements based on the feedback provided. Their revisions have strengthened the overall quality and clarity of the work. I believe the manuscript now meets the journal’s standards and recommend it for publication.

Reviewer #4: (No Response)

.

Reviewer #2: **Yes:** Mohamed AbdellatiefMohamed AbdellatiefMohamed AbdellatiefMohamed Abdellatief

Reviewer #4: No

---

## [Editor Report · Acceptance letter]

PONE-D-25-23864R1

PLOS One

Dear Dr. Jitang,

I'm pleased to inform you that your manuscript has been deemed suitable for publication in PLOS One. Congratulations! Your manuscript is now being handed over to our production team.

Kind regards,

on behalf of

Dr. Kaywan Othman Ahmed

Academic Editor

PLOS One